# The Extract of *Corydalis yanhusuo* Prevents Morphine Tolerance and Dependence

**DOI:** 10.3390/ph14101034

**Published:** 2021-10-12

**Authors:** Lamees Alhassen, Khawla Nuseir, Allyssa Ha, Warren Phan, Ilias Marmouzi, Shalini Shah, Olivier Civelli

**Affiliations:** 1Department of Pharmaceutical Sciences, School of Pharmacy, University of California-Irvine, Irvine, CA 92697, USA; lalhasse@uci.edu (L.A.); Knuseir@uci.edu (K.N.); allyssch@uci.edu (A.H.); wtphan@uci.edu (W.P.); ilias.marmouzi@gmail.com (I.M.); 2Department of Clinical Pharmacy, Jordan University of Science and Technology, Irbid 22110, Jordan; 3Department of Anesthesiology and Perioperative Care, School of Medicine, University of California-Irvine, Irvine, CA 92697, USA; ssshah1@hs.uci.edu; 4Department of Developmental and Cell Biology, School of Biological Sciences, University of California-Irvine, Irvine, CA 92697, USA

**Keywords:** opioid epidemic, addiction, dependence, medicinal plant, traditional medicine, *Corydalis yanhusuo*, antinociception

## Abstract

The opioid epidemic was triggered by an overprescription of opioid analgesics. In the treatment of chronic pain, repeated opioid administrations are required which ultimately lead to tolerance, physical dependence, and addiction. A possible way to overcome this conundrum consists of a co-medication that maintains the analgesic benefits of opioids while preventing their adverse liabilities. YHS, the extract of the plant *Corydalis yanhusuo*, has been used as analgesic in traditional Chinese medicine for centuries. More recently, it has been shown to promote analgesia in animal models of acute, inflammatory, and neuropathic pain. It acts, at least in part, by inhibiting the dopamine D2 receptor, suggesting that it may be advantageous to manage addiction. We first show that, in animals, YHS can increase the efficacy of morphine antinociceptive and, as such, decrease the need of the opioid. We then show that YHS, when coadministered with morphine, inhibits morphine tolerance, dependence, and addiction. Finally, we show that, in animals treated for several days with morphine, YHS can reverse morphine dependence and addiction. Together, these data indicate that YHS may be useful as a co-medication in morphine therapies to limit adverse morphine effects. Because YHS is readily available and safe, it may have an immediate positive impact to curb the opioid epidemic.

## 1. Introduction

Over the past two decades, dramatic increases in opioid overdose mortality have occurred in the United States and other nations. Recognized as a public health crisis, it is commonly referred to as the opioid epidemic [1]. The opioid epidemic started with an increase in opioid prescriptions to treat chronic pain. Chronic pain is a therapeutic challenge and its management by opioids is controversial [2]. Non-opioid treatments should be the preferred first step, but are often replaced by opioid medications as conditions worsen [3].

Opioids are administered for their superior analgesic effectiveness. However, associated with repetitive opioid administration is the development of tolerance, which represents a loss of efficacy upon time [4]. Pain-afflicted patients require higher doses of opioids to maintain a mostly pain-free state, which in turn increases the risk of dependence, addiction, and fatal overdose [5,6]. In animals, tolerance is monitored by quantifying analgesic responses over repeated opioid administration [7]. Repeated administration of opioids also leads to physical dependence, i.e., the need for maintained administration. Dependence manifests itself with the emergence of withdrawal symptoms when the use of opioids is abruptly discontinued and can be precipitated by opioid antagonists [7,8,9]. It can be quantified by treating opioid-dependent animals with naloxone and monitoring withdrawal symptoms. Dependence is also associated with the desire to repeat the positive reinforcing effects of the opioids [10]. While it involves numerous neuronal systems, it is predominantly mediated by the dopaminergic mesocorticolimbic reward system [10,11,12,13]. In experimental animals, addiction can be assessed by using assays monitoring drug-seeking behaviors, such as conditioned place preference (CPP) [14,15,16].

The US Centers for Disease Control and Prevention (CDC) has issued guidelines aiming at reducing the use of opioid drugs [1,17]. Recommendations include prescribing over-the-counter pain relievers, such as acetaminophen and ibuprofen in lieu of opioids. Nonsteroidal anti-inflammatory drugs (NSAIDs) are most effective against mild to moderate pain linked to inflammation. Indeed, the lack of efficacy of acetaminophen in chronic pain conditions has been documented [18,19,20].

The CDC guidelines also accept the use of opioids in combination with non-opioid therapy after careful assessment of pain control. We reflected that, ideally, the co-administration of a non-opioid drug should not only decrease the need of the opioid drug but also prevent its tolerance-inducing or addictive properties [3]. The extract of the plant *Corydalis yanhusuo* (YHS) may offer such an opportunity as a safe and readily available co-medication in the treatment of chronic or severe pain.

YHS has been used as an analgesic in traditional Chinese medicine (TCM) for centuries [21,22]. We have reported that YHS effectively attenuates acute, inflammatory and chronic pain in animal models. It elicits these responses without causing tolerance. Its mode of action relies at least in part on its antagonistic activity at the dopamine D2 receptor [21]. This suggested to us that it may also have anti-addictive properties. Therefore, we evaluated the antinociceptive activity of YHS in combination with morphine as well as its role in decreasing morphine tolerance and dependence. We finally assess YHS abilities to reverse these responses.

## 2. Results

The antinociceptive activity of morphine in the presence of YHS.

Antinociceptive activities were measured using foot withdrawal latency (FWL) in the hot plate assay [23]. Mice were first tested before injection for their basal FWL response which was found to be similar in the range of 3–7 secs. Then, mice were injected intraperitoneally (i.p.) with saline or variable doses of morphine (M) or YHS (Y). FWL was measured at 30 min, 60 min, and 120 min after the injection. Figure 1A shows the FWL of mice injected with saline or morphine (2.5 mg/kg), YHS (250 mg/kg), or the combination of morphine and YHS (2.5 mg and 250 mg/kg, respectively) at different times after injection. At these doses, morphine induces a FWL response that is smaller than YHS but when both are combined the response is strongly increased. When variable doses of morphine, YHS, or YHS and morphine are compared (Figure 1B), dose responses show that YHS potentiates the antinociceptive activity of morphine. Indeed, co-administration of YHS at 250 mg/kg increases the morphine 2.5 mg/kg FWL response to that equivalent to morphine 10 mg/kg.

Tolerance was assessed by injecting morphine, YHS, or a combination of both twice daily for 6 days. As shown in Figure 2A and Appendix A, morphine (2.5 mg/kg) lost its antinociceptive activity over the seven-day period while analgesic effects of YHS (250 mg/kg) were retained. Moreover, when combined, YHS was able to completely prevent tolerance development to any morphine dose tested, while retaining the additive antinociceptive effect of the combination (Figure 2B).

To investigate the effects of YHS and morphine on withdrawal, we observed chronically treated animals after naloxone injection. Withdrawal behaviors, such as body grooming, jumping, writhing, head shakes, genital licking, face wiping, teeth chattering, dysphoria, rearing, chewing, and diarrhea, were scored over a 15-min time period. Figure 3 displays the various withdrawal behaviors observed in each group. While morphine-treated mice (2.5 and 5 mg/kg) showed significantly elevated signs of nalone-precipitated withdrawal, animals chronically treated with YHS (250 mg/kg) or a combination of morphine (2.5 mg/kg) + YHS displayed no withdrawal behaviors and were indistinguishable from saline-treated controls. It appears that co-administration of YHS (250 mg/kg) is sufficient to prevent establishment of typical morphine dependence. Sniffing was unchanged in all treatment groups and served as an internal control for natural behavior.

Reward-related behavior was monitored in the CPP paradigm. A preference score was determined by observing the time that each animal spent in the drug-paired chamber vs. the saline-paired chamber during the pre- and post-training periods. As shown in Figure 4, M2.5 and M5 induced strong preference scores (100 and 250, respectively), indicative of addiction and the associated drug-seeking. When YHS 250 was added to M2.5 or M5 during the training, preference scores decreased strongly, showing that YHS is able to significantly limit the rewarding effects of morphine.

The previous data suggest that co-administration of morphine and YHS could prevent development of tolerance and dependence in morphine-naive animals. We next investigated whether YHS might also be beneficial in already-dependent animals and could, thus, potentially be used to curb the opioid epidemic. For tolerance reversal, mice were treated for 3 days with morphine (2.5 mg/kg twice daily), followed by four days of YHS (250 mg/kg) or YHS + morphine (250 mg/kg and 2.5 mg/kg, respectively) and then tested for analgesic tolerance (Figure 5). YHS alone or in combination with morphine reverses morphine tolerance and restores analgesia in previously tolerant animals.

The same animals were also tested for naloxone-precipitated withdrawal (Appendix A). As expected, mice treated with morphine at 2.5 mg/kg for seven days exhibited significant signs of withdrawal behaviors (Appendix A). Switching treatment from morphine alone to morphine 2.5 mg/kg + YHS 250 mg/kg completely prevented—and may indeed have reversed—opioid dependence. Similarly, animals treated with morphine 2.5 mg/kg for 3 days and then switched to YHS also showed no signs of persisting morphine dependence, although we cannot rule out the possibility that these mice already underwent morphine withdrawal during the four days of YHS treatment. Importantly, adding YHS to an ongoing morphine treatment regimen appears to reverse pre-existing morphine dependence while maintaining high antinociceptive efficacy. For reversal of morphine rewarding properties, mice were either treated with saline or morphine (2.5 mg/kg) for 7 days. After this initial treatment, mice were kept on saline or saline treatment, or switched to YHS (250 mg/kg) or morphine (2.5 mg/kg) + YHS (250 mg/kg) for another 7 days. Conditioned place preference was tested to observe for any addiction-like properties. Mice that were treated initially with morphine and then switched to either YHS (250 mg/kg) or morphine (2.5 mg/kg) + YHS (250 mg/kg) reversed any addiction-like behavior in mice (Figure 6).

## 3. Discussion

The opioid epidemic resulted in an increase in opioid prescriptions to treat neuropathic and other forms of chronic pain. Opioids are effective for treating severe acute pain, but display less effectiveness in treating chronic pain. Chronic pain requires repetitive administration of an antinociceptive agent. When opioids are administered repeatedly, they induce analgesic tolerance, which is a time-dependent loss of efficacy requiring increasing doses to reach the same antinociceptive state [1,24,25]. Repeated administrations of opioids then lead to physical dependence, i.e., the need for maintained administration to prevent withdrawal symptoms. Dependence manifests itself in withdrawal when the use of opioids is abruptly discontinued [2,26] and is accompanied by addictive behaviors to re-experience the rewarding effects of the opioids or at least avoid withdrawal [27,28,29].

The opioid epidemic stems from pain-sufferers becoming addicts because they have to simultaneously cope with pain, opioid tolerance, and dependence. To fight the epidemic, the CDC has recommended using NSAIDS alone or in combination with opioid drugs to reduce their use. However, NSAIDs often lack efficacy against neuropathic pain [3,18]. A safer pain medication would be one that could limit or inhibit all three opioid-related phenomena: tolerance, dependence, and addiction [1,20,30]. In the present study, we present evidence that the extract of the plant *Corydalis yanhusuo* (YHS) may be able to maintain the analgesic benefits of opioids while curbing their adverse liabilities when administered as a co-medication.

We first show that adding YHS to morphine potentiates its analgesic activity in a dose- and time-dependent manner. While a single low dose of morphine (2.5 or 5 mg/kg) induces analgesia 30 min after its administration but begins to display tolerance 30 min later, YHS alone or morphine with YHS retain most of their antinociceptive activities for over a 2 h period. When co-administered, YHS at all doses tested increases the antinociceptive effect of morphine, showing that it could serve as an adjuvant to opiates in managing pain. This would allow for lowering the doses of morphine at the same level of antinociceptive effectiveness, thus decreasing the risk of addiction.

An important clinical limitation of opioid treatment is the development of tolerance. Over repeated administrations, opioids lose their potencies [10,28,31]. We assessed the effect of adding YHS to morphine on tolerance. While morphine induces definitive tolerance over 7 days of repeated administration, the addition of YHS prevents morphine tolerance. This implies that the need to increase morphine doses to maintain antinociception would not occur in the presence of YHS, a factor that could significantly help to reduce addiction. Moreover, we show that co-administration of YHS with morphine can prevent the establishment of opioid dependence in drug-naive animals. More clinically relevant is the observation that a combination of YHS and morphine can reverse a preexisting morphine dependence and could thus help addicted patients to escape the vicious cycle of continued opioid exposure.

In trying to understand the mechanism underlying YHS mode of action, we have to recognize that YHS is a complex extract and that its activity may rely on several components. YHS contains over 100 chemical components [27]. Of these, some 81 are protoberberine, apomorphine, opiate-like, and other alkaloids [32,33]. Several YHS components have been pharmacologically analyzed and found to display antagonism at dopamine receptors [21], agonism at opioid receptors [34], or inhibition of acetylcholinesterase activity [35]. These components act at their respective receptors at high nanomolar or micromolar concentrations. Of the YHS components, possibly the most studied is L-tetrahydropalmatine (L-THP) [36]. L-THP has been shown to exhibit sedative, anti-epileptic, antidepressant, and anxiolytic effects in addition to its analgesic activity [36]. With regard to drugs of abuse, L-THP has been shown to attenuate cocaine-associated reward [36], self-administer cocaine, and promote recovery from cocaine-induced effects in rats [36,37]. L-THP was used to treat heroin withdrawal syndrome in patients [36]. L-THP was found to significantly reduce heroin craving and withdrawal symptoms and to improve the abstinence rate of heroin addicts. In these experiments L-THP was given at a dose of 60 mg twice daily. L-THP represents approximately 0.2% of the total dry mass of YHS [36], which is usually taken at about 5–10 g per day. L-THP should therefore not account for the full efficacy of YHS. Dehydrocorybulbine (DHCB), another YHS component and chemically related to L-THP, may also partially contribute to YHS effects in models of drug addiction or analgesia [36]. However, it has been shown that a combination of L-THP and DHCB does not account for the entire YHS analgesic effect [38]. It is therefore probable that several of the YHS components may participate in its efficacy and that YHS polypharmacological profiles are required for its full efficacy against morphine tolerance and dependence. Alternatively, undiscovered YHS component(s) may be responsible for these beneficial effects. Further studies will be needed to fully understand YHS mode of action.

## 4. Materials and Methods

### 4.1. Animals

CD-10 male mice obtained from Charles River aged 8 weeks were used for all behavior experiments. Mice were group housed and maintained on a 12 h light/dark cycle with food and water available ad libitum. All behaviors and treatments are approved by the Institutional Animal Care and Use Committee of the University of California, Irvine (AUP #20-015). Animals were frequently weighed and observed after injections to ensure proper health and weight gain. Any animals that seemed to be in distress or abnormal were excluded from the study.

### 4.2. Drugs

#### 4.2.1. Morphine

Morphine injectable C II obtained from Patterson Veterinary (10 mg/mL concentration) (07-892-4699) was used according to the assigned groups and behavioral experiment. Three doses were used: 10 mg/kg, 5 mg/kg, and 2.5 mg/kg. Morphine was injected twice daily (morning and afternoon) for seven days intraperitoneally (i.p.) in a volume of 5 uL/g. The dosage for each animal was determined based on the animal’s body weight. Morning and afternoon injections were spaced out for about 10 h. The varied doses of morphine used was dissolved in sterile saline. The vehicle used for all experiments was sterile saline.

#### 4.2.2. Naloxone

Naloxone hydrochloride dihydrate obtained from Sigma Aldrich (N7758) at a dose of 6 mg/kg was given to precipitate withdrawal-like symptoms in mice at the end of the hot plate assay. The dosage for each animal was determined based on the animal’s body weight in a volume of 5 uL/g. Naloxone was injected via intraperitoneal administration.

#### 4.2.3. YHS

The *Corydalis yanhusuo* extract (YHS) obtained from Dongyang County (Zhejiang, China) and authenticated by Institute of Medication, Xiyuan Hospital of China Academy of Traditional Chinese Medicine, as previously described [21]. A dose of 250 mg/kg dose [21] in a volume of 5 uL/g was used for most assays. To demonstrate a dose response of YHS, both the 125-mg/kg and the 500-mg/kg doses were used. The YHS powder was dissolved in saline solution and injected via intraperitoneal administration.

#### 4.2.4. Animal Groups and Treatments

Mice were divided into various groups: control (saline only), YHS (125, 250, 500 mg/kg), morphine (10, 5, 2.5 mg/kg), and the combination of YHS and morphine at varying YHS concentrations (125, 250, 500 mg/kg). Drugs or vehicle were injected twice daily for seven days in the hot plate assay, and twice daily for 6 days in the conditioned place preference assay (CPP). The same mice are used for the pain, tolerance, and withdrawal assay. Behavioral testing was performed according to each model explained below.

### 4.3. Behavioral Assays

#### 4.3.1. Pain Model

To establish the antinociceptive effects of YHS and morphine in combination on acute pain, foot withdrawal latency (FWL) in the hot plate assay was measured 30, 60, and 120 min after injections, as well as before to establish a baseline on day 1 at 52 degrees Celsius. A hotplate assay was performed, as described in the literature [22,23]. The cutoff time for the hotplate assay was 50 s.

#### 4.3.2. Tolerance Assay

Mice were injected twice daily and the hot plate assay was tested on day 7 for tolerance-like behavior. FWL on the hotplate was measured at 30, 60, and 120 min after injections. Day 1 and Day 7 responses were compared to observe any tolerance to the drugs.

#### 4.3.3. Withdrawal Assay

The withdrawal assay was tested on day 8 for withdrawal-like behaviors. Mice were observed for withdrawal symptoms, such as teeth chattering, genital licking, face wiping, head shakes, etc. [7]. The total time for this assay was about 50 min. All sessions were videotaped and analyzed later by individuals blinded to the experiment. Mice were habituated to a 40-x 40 locomotor test chamber for 5 min. After 5 min, the mice were injected according to their previous treatment (YHS, Morphine, Saline, or YHS+M2.5). Mice were observed for 30 min after injection and then were injected with naloxone followed by observation for 15 min.

#### 4.3.4. Conditioned Place Preference (CPP)

The CPP assay consisted of multiple stages: a habituation period, conditioning sessions, and a test day as previously described [24]. The CPP assay consisted of a three chamber box. The middle chamber is considered the neutral side, where the mouse is placed and is allowed to explore the other two sides of the chamber. One side of the chamber is decorated with stripes and the other side with circles. Mice were then habituated to all three compartments of the chamber for 3 days for 10 min. Animals that display a preference during the habituation period are omitted from the study. After the habituation period was conducted, mice were conditioned to both sides of the chamber (morning and afternoon) for 7 days and were injected with their corresponding drug (i.e., morphine group animals are injected with saline on the circles side in the morning and then morphine on the stripes at night). After the conditioning sessions, mice were given a day in between with no injections to induce drug craving, as previously described [16]. The next day, mice were tested for their preference in a total duration of 10 min. Preference score was calculated by using the equation: ((A2 − A1)/A1) ∗ 100, where A2 represents the percent time spent on the most preferred side during the final preference test and A1 represents the percent time spent on this same side during the initial habituation period.

To test CPP in mice dependent on morphine, animals were first treated with morphine and then switched to a treatment of YHS or the combination of M2.5 and YHS. This experiment was conducted the same way as described above, except it includes 7 days of morphine conditioning, and then an additional 7 days of the alternative treatment (either YHS or M2.5 + YHS). Once again, the mice were given a day with no injections before testing their CPP the next day. The groups that started with morphine were never taken off it to avoid any withdrawal symptoms that may arise and interfere with analyzing addiction-like behavior. All CPP testing sessions were videotaped and analyzed by individuals blinded by the experiment. A preference score for each mouse was calculated based on the equation described above.

### 4.4. Statistical Analysis

GraphPad Prism (GraphPad Software, Inc., San Diego, CA, USA) was used for statistical analysis, and all data are presented as mean ± standard error mean (SEM). One-way ANOVA followed by Tukey post-test was used to analyze morphine antinociception, dependence, morphine-induced CPP, reversal of CPP, and reversal of dependence. Two-way ANOVA followed by multiple comparisons tests was used to analyze inhibition of morphine tolerance and reversal of morphine tolerance. *p* value < 0.05 was deemed statistically significant.

## 5. Conclusions

In summary, although therapeutic plants were studied for many years for various ailments, the combination of opioids and herbal medicine for the management of pain and addiction has not been effectively explored. We show that the extract of the plant *Corydalis yanhusuo* (YHS) is able to reduce required doses of morphine in pain management and can successfully block development of morphine tolerance and dependence. Moreover, YHS is able to reverse a previously established opioid dependence. YHS therefore displays advantageous properties in our aim to curb the opioid epidemic. The fact that it is safe and readily available implies that it could have an immediate effect on this epidemic and that clinical trials are warranted.

## Figures and Tables

**Figure 1 pharmaceuticals-14-01034-f001:**
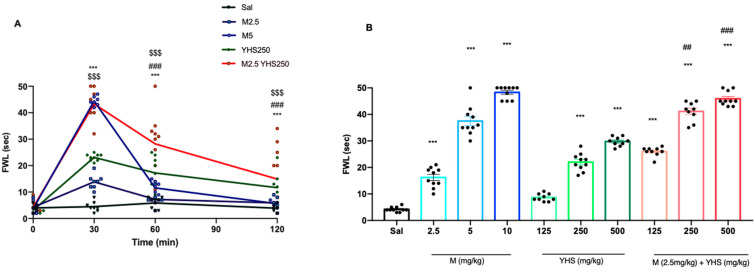
YHS increases morphine antinociception. **A:** Foot withdrawal latency (FWL) of mice injected with saline, morphine (2.5 mg/kg), YHS (250 mg/kg), or the combination of morphine and YHS (2.5 mg and 250 mg/kg, respectively) at 30, 60, and 120 min after i.p. administration (n = 10). The black dots correlate to the number of animals used in each experiment. Two-way ANOVA revealed significant drug effects F (3, 288) = 332.8 *p* < 0.0001, time effect F (4, 288) = 217.5 *p* < 0.0001, and drug x interaction time F (12, 288) = 55.29 *p* < 0.0001, followed by Tukey’s multiple comparison test *** *p* < 0.001 compared with M2.5 mg/kg, $$$ *p* < 0.0001 compared with saline, ### *p* < 0.0001 compared with M5 mg/kg. **B:** FWL at 30 min after morphine (2.5 mg/kg, 5 mg/kg, 10 mg/kg), YHS (125 mg/kg, 250 mg/kg, 500 mg/kg),or morphine + YHS (n = 9–10) i.p. administration. One-way ANOVA revealed significant drug effects F = 247.2, *p* < 0.0001, followed by Tukey’s multiple comparison test *** *p* < 0.001 compared with saline, ## *p* < 0.01, ### *p* < 0.001 compared with M2.5 mg/kg. YHS prevents morphine tolerance.

**Figure 2 pharmaceuticals-14-01034-f002:**
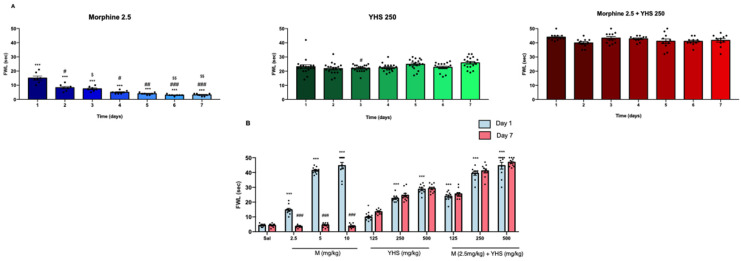
YHS inhibits morphine tolerance. **A:** FWL at 30 min after morphine 2.5 mg/kg (M2.5), YHS (250 mg/kg), and morphine +YHS (M2.5+YHS) (i.p. administration) over a period of 7 days (n = 7–11) to display tolerance. The black dots correlate to the number of animals used in each experiment. The gradient color for each figure shows the time-dependent change. One-way ANOVA followed by Tukey’s test revealed significant drug tolerance over 7 days F = 33.59, *p* < 0.0001, D1 *** *p* < 0.001 compared with D2-D7, D2 ### *p* < 0.0001, ## *p* < 0.01 compared with D4-D7, D3 $$ *p* < 0.01, $ *p* < 0.05 compared with D6 and D7 in the morphine (2.5 mg/kg) group. One-way ANOVA followed by Tukey’s test revealed no significant drug tolerance over 7 days for YHS (250 mg/kg) group, D3 # *p* < 0.05 compared with D7. One-way ANOVA followed by Tukey’s test revealed no significant drug tolerance over 7 days for the combination group. **B:** Comparison of FWL at day 1 and day 7 30 min after i.p. administration of morphine, YHS, and morphine + YHS (n = 10). Two-way ANOVA revealed significant tolerance amongst all morphine doses (2.5, 5, 10 mg/kg) F(9, 180) = 291.3 *p* < 0.0001, followed by Tukey’s multiple comparison test, ### *p* < 0.001. Two-way ANOVA revealed significant analgesic effects between saline and all other groups on Day 1, F(9, 180) = 291.3 followed by Tukey’s multiple comparison test *** *p* < 0.001 compared with saline, *p* < 0.05 compared with saline. YHS prevents morphine dependence.

**Figure 3 pharmaceuticals-14-01034-f003:**
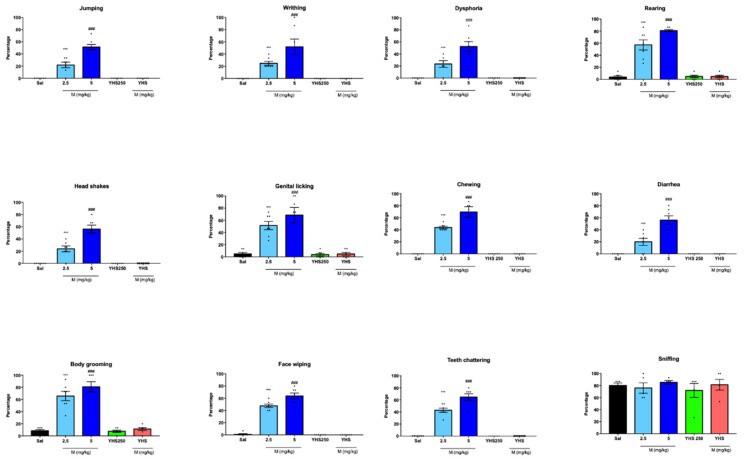
YHS inhibits morphine physical dependence after i.p. administration. Jumping, writhing, head shakes, genital licking, body grooming, face wiping, teeth chattering, dysphoria, rearing, chewing, diarrhea, and sniffing after naloxone injection (n = 8). One-way ANOVA revealed significant percentage of jumping, writhing, head shakes, genital licking, body grooming, face wiping, teeth chattering, dysphoria, rearing, chewing, and diarrhea in both morphine groups, F = 65.51 *p* < 0.0001, followed by Tukey’s multiple comparison test M2.5 *** *p* < 0.0001 compared with saline, M5, YHS 250, and M2.5YHS250; M5 ### *p* < 0.0001 compared with saline, M2.5, YHS 250, and M2.5 YHS250. YHS inhibits the rewarding properties of morphine.

**Figure 4 pharmaceuticals-14-01034-f004:**
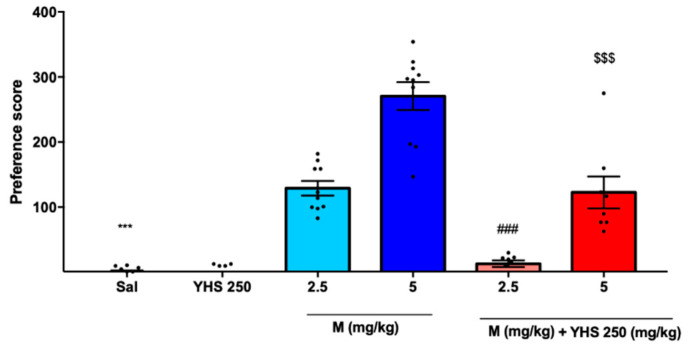
Morphine-induced CPP is inhibited by YHS after i.p. administration. Preference scores calculated based on animal’s time spent in the drug-paired chamber vs. the non-drug-paired chamber during the pre- and post-preference periods (n = 8–11). The black dots correlate to the number of animals used in each experiment. One-way ANOVA revealed significant drug addiction in all morphine groups and a reduction in the combination groups F = 62.50 *p* < 0.0001, followed by Tukey’s multiple comparison test, *** *p* < 0.0001 compared with M2.5, M5, and M5 YHS250, ### *p* < 0.0001 compared with M2.5, M5, and M5 YHS250, $$$ *p* < 0.0001 compared with M5 and M2.5 YHS 250. YHS reverses morphine tolerance and dependence.

**Figure 5 pharmaceuticals-14-01034-f005:**
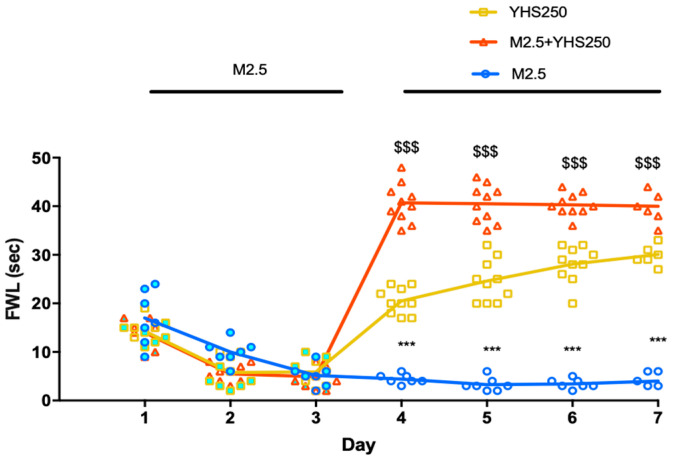
YHS reverses morphine tolerance. Analgesic response after 3 days (D) of Morphine, followed by 4 D of either YHS (250 mg/kg) or M2.5-M2.5+YHS (n = 10) (i.p. administration). Two-way ANOVA revealed significant drug effects F (6, 168) = 205.9 *p* < 0.0001, followed by Tukey’s multiple comparison test, *** *p* < 0.0001 M2.5 compared with M2.5-YHS and M2.5-M2.5-YHS on D4-7, $$$ *p* < 0.0001 compared M2.5-YHS with M2.5-M2.5-YHS on D4-7.

**Figure 6 pharmaceuticals-14-01034-f006:**
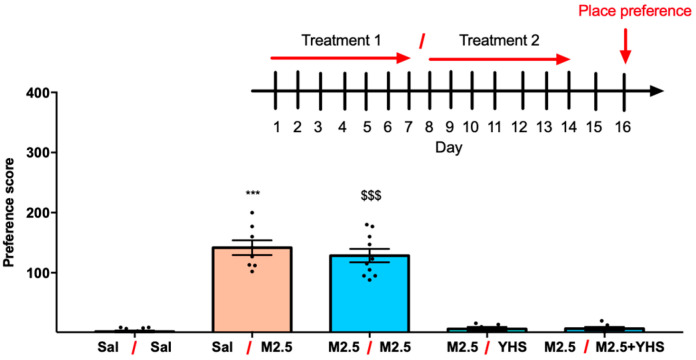
YHS reverses morphine-induced CPP. CPP responses for the following groups: 14 days of Sal injections (Sal–Sal), 7 days of Sal followed by 7 days of M2.5 injections (Sal-M2.5), 14 days of M2.5 injections (M2.5-M2.5), 7 days of M2.5 followed by 7 days of YHS (250 mg/kg) or M2.5-YHS injections (i.p. administration). (n = 7–11). The black dots correlate to the number of animals used in each experiment. One-way ANOVA revealed significant drug preference F = 8.131 *p* < 0.0001, followed by Tukey’s multiple comparison test, *** *p* < 0.0001 compared with Sal and M2.5-YHS, $$$ *p* < 0.0001 compared with Sal-Sal, M2.5-YHS, and M2.5-M2.5+YHS.

## Data Availability

All data is contained within the article and Appendix A.

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
