# Peer review of "The Extract of Corydalis yanhusuo Prevents Morphine Tolerance and Dependence"

_pharmaceuticals, 2021, doi:10.3390/ph14101034_

Round 1

Reviewer 1 Report

The manuscript by Alhassen et al. presents a pharmacological study on an extract of the medicinal plant Corydalis yanhusuo (YHS) in combination with morphine in relation to antinociception and development of morphine tolerance and dependence. Although the topic of the manuscript is of significant relevance in the light of the current opioid epidemic, and the results are interesting, there were several major issues that need to be addressed by the authors.

A major critical point to the manuscript is that the authors refer throughput the manuscript to chronic pain, but in their study, they use a model of acute thermal pain (hot-plate test). Why they have not evaluated if the combination of YHS and morphine would also have the same beneficial effects in a model of chronic pain (inflammatory or even neuropathic)?

Did the authors’ test/observe other changes in the condition of the animals (any side effects i.e. sedation, constipation, respiratory depression, locomotor activity) induced by the different treatments, which may be important, particularly in the context of pain management.

The main question is still open on the mechanism of YHS antinocicieptive activity and how it prevents morphine tolerance and dependence?

Other comments:

Title: “Corydalis yanhusuo” should be added to the title.

Revise to Corydalis yanhusuo throughout the manuscript.

Lines 13-14: Is this a required statement on the study?

Keywords: revised to “Corydalis yanhusuo” and “antinociception”

All figures: The i.p. route of administration of drugs should be specified.

Figure 1. Suppl. Should this be included in the main text?

Discussion: lines 245 and 272: Replace “potency” by “effectiveness”.

Line 246: Revise to “repetitive administration”.

Line 309: Give body weight of mice.

Lines 325-328: How was naloxone administered, and in which volume?

Line 334: Was YHS powder truly and fully soluble in saline solution? In another article by the same lab (DOI:10.1371/journal.pone.0162875), YHS was dissolved in a vehicle solution of cremophor EL: ethanol: saline (2:1:17). This needs to be confirmed for the accuracy of the information in the present manuscript.

Line 357: Revised to “was about 50 min”.

4.3.3. The method description of the withdrawal assay on page 10 is largely different from the procedure stated on page 9 (4.2.4). This needs to be checked for the accuracy of information. Furthermore, withdrawal signs are commonly assessed two hours after last drug injection and not one hour as in the current study.  

4.3.4. CPP – change the tense in the first paragraph (lines 365-380) from present to past tense.

It should be specify how long after drug administration this test was performed.

Author Response

Title: “Corydalis yanhusuo” should be added to the title.

Since both reviewers asked for Corydalis yanhusuo to be included in the title we modified the title accordingly.

Revise to Corydalis yanhusuo throughout the manuscript.

We have revised this in the manuscript. 

Lines 13-14: Is this a required statement on the study?

We have kept this in the manuscript. 

Keywords: revised to “Corydalis yanhusuo” and “antinociception”

We have revised this in the manuscript. 

All figures: The i.p. route of administration of drugs should be specified.

We have revised this in the manuscript. 

Figure 1. Suppl. Should this be included in the main text?

Figure 1 Suppl was presented as part of the text. We think that it would be better put in supplements. We modified Fig 2 to better describe the tolerance effect and added the statistical analysis of this figure in supplements (Suppl. 1)

Discussion: lines 245 and 272: Replace “potency” by “effectiveness”.

We have revised this in the manuscript.

Line 246: Revise to “repetitive administration”.

We have revised this in the manuscript. 

Line 309: Give body weight of mice.

We have revised this in the manuscript.

Lines 325-328: How was naloxone administered, and in which volume?

We have revised this in the manuscript.

Line 334: Was YHS powder truly and fully soluble in saline solution? In another article by the same lab (DOI:10.1371/journal.pone.0162875), YHS was dissolved in a vehicle solution of cremophor EL: ethanol: saline (2:1:17). This needs to be confirmed for the accuracy of the information in the present manuscript.

We have found that YHS when dissolved in cremophor EL: ethanol: saline has the same effect as when dissolved in saline alone. Because of the potential use of YHS in human trials we carried our study with YHS dissolved in saline

Line 357: Revised to “was about 50 min”.

We have revised this in the manuscript. 

4.3.3. The method description of the withdrawal assay on page 10 is largely different from the procedure stated on page 9 (4.2.4). This needs to be checked for the accuracy of information. Furthermore, withdrawal signs are commonly assessed two hours after the last drug injection and not one hour as in the current study.  

We have revised this in the manuscript. The same mice used in the tolerance assay were also used in the withdrawal assay. We have used the protocol (cited as reference 7) to confirm the withdrawal assay protocol and timing after injection.

4.3.4. CPP – change the tense in the first paragraph (lines 365-380) from present to past tense.

We have revised this in the manuscript. 

It should be specify how long after drug administration this test was performed.

We have revised this in the manuscript.

Reviewer 2 Report

In this paper, the effects Corydalis yanhusuo extract (YHS) on morphine-induced antinociception, tolerance and dependence was studied in mice. YHS increase the antinociceptive effect induced by morphine and YHS inhibited morphine tolerance, dependence and addiction. YHS also reversed morphine dependence and addiction. The authors concluded that YHS might be useful as a co-medication in morphine therapies to limit adverse morphine effects.

I think this manuscript has merits, but the authors should consider the following comments and suggestion before publication.

  • The authors should report the name of the plant used in their experiments in the paper title.
  • Page 2, lines 82-89 and figure 1. Since the doses of YHS used induce an antinociceptive effect, it seems to me that the effects of morphine and YHS add up synergistically. In my opinion, the authors should have used ineffective doses of YHS to test for a potentiating effect induced by YHS on the effects of morphine.
  • Are there data in the literature on the possible effects induced by YHS on the pharmacokinetics of morphine?
  • How do the authors explain the lack of tolerance induced by YHS treatment?
  • Authors should briefly describe the procedures used to obtain the YHS extract.
  • Authors should describe how they measured the distress caused by the treatments.
  • Page 9, line 321. Please, check the volume utilized for morphine administration. In addition, authors should indicate the volume of administration used for other treatments.

Author Response

The authors should report the name of the plant used in their experiments in the paper title.

Since both reviewers asked for Corydalis yanhusuo to be included in the title we modified the title accordingly

Page 2, lines 82-89 and figure 1. Since the doses of YHS used induce an antinociceptive effect, it seems to me that the effects of morphine and YHS add up synergistically. In my opinion, the authors should have used ineffective doses of YHS to test for a potentiating effect induced by YHS on the effects of morphine.

We believe that our data show that YHS has an additive effect on morphine (Fig 1B). In our previous manuscript (Wang et al 2016) we show that YHS at low doses has a weak but still measureable antinociceptive effect. Using ineffective doses of YHS should not result in a potentiating effect on morphine.

Are there data in the literature on the possible effects induced by YHS on the pharmacokinetics of morphine?

We have not found any such data in the literature

How do the authors explain the lack of tolerance induced by YHS treatment?

We believe that YHS acts on receptor(s) that prevent not only its own but also morphine tolerance. This is part of our present hypothesis and we are looking into it. 

Authors should briefly describe the procedures used to obtain the YHS extract.

We describe in our previous manuscript how the YHS was obtain. We have added this is the methods

Authors should describe how they measured the distress caused by the treatments.

There was no evident distress when YHS or morphine were administered. In the dependence assays we show that animals display several aspects of distress upon naloxone administration. None of these were detected without naloxone administration

Page 9, line 321. Please, check the volume utilized for morphine administration. In addition, authors should indicate the volume of administration used for other treatments.

We have revised this in the manuscript. 

Round 2

Reviewer 1 Report

The authors have addressed all my concerns.